# The Role of Periodic Structures in Light Harvesting

**DOI:** 10.3390/plants10091967

**Published:** 2021-09-20

**Authors:** Eugene Bukhanov, Alexandr V. Shabanov, Mikhail N. Volochaev, Svetlana A. Pyatina

**Affiliations:** 1Kirensky Institute of Physics FRC «KSC of SB RAS», Academgorodok str. 50/12, 660036 Krasnoyarsk, Russia; alexch_syb@mail.ru (A.V.S.); volochaev91@mail.ru (M.N.V.); 2Federal Research Center «KSC of SB RAS», Academgorodok str. 50, 660036 Krasnoyarsk, Russia; davcbetik@mail.ru; 3Institute of Fundamental Biology and Biotechnology, Siberian Federal University, 79 Svobodnyi av., 660041 Krasnoyarsk, Russia

**Keywords:** chloroplast, thylakoid, chlorophyll, light harvesting, density of photon states, photosynthesis, biophotonic crystal

## Abstract

The features of light propagation in plant leaves depend on the long-period ordering in chloroplasts and the spectral characteristics of pigments. This work demonstrates a method of determining the hidden ordered structure. Transmission spectra have been determined using transfer matrix method. A band gap was found in the visible spectral range. The effective refractive index and dispersion in the absorption spectrum area of chlorophyll were taken into account to show that the density of photon states increases, while the spectrum shifts towards the wavelength range of effective photosynthesis.

## 1. Introduction

Photosynthesis is surely one of the most important processes that occurs in a green leaf under the influence of light. It is considered to be multi-stage. At the first stages the light is absorbed, and then the charge is transferred in thylakoids membranes within chloroplasts. An important role in absorption of light is played by pigment complexes containing chlorophylls, carotenoids, and proteins [1]. The formation of plant pigments, their effect on color, and efficiency of photosynthesis have been studied for a long period of time [1,2,3]. Nevertheless, the importance of structural coloration has only recently begun to be taken into consideration [4,5,6,7,8], thanks to new methods that use high resolution to examine the structure [4]. Structural coloration is thought of as resonant reflection or transmission that emerges due to the ordering of objects. We know that reflection ranges of certain wavelengths appear in the spectrum of structures that are formed of regular ordered arrays of submicron scale. The mechanisms letting plants, algae, and bacteria generate fairly regular nanoscale structures are not yet fully explained. However, some remarkable works take into account the cellular genetic basis of the structural coloration of plants [9] and antenna proteins functions [10], as well as lattice models of organization, in stacks of thylakoids membranes [11]. There are several forms of structural color that are developed by surface diffraction gratings, multilayer reflectors, and helicoidal formations. Surface diffraction gratings form nanoscale ridge patterns. A multilayer reflector is a stack of alternating layers, each having its own thicknesses and refractive indices.

Different anisotropic formations present in a plant at different scales [12]. For instance, at molecular level, layers can consist of water, cellulose, air, proteins, and lipids. Taken together, these layers form wider periodic structures comparable to the wavelength of light. Such multilayer systems are usually called biophotonic crystal systems [4]. They got their name similarly to photonic crystals—artificially created superlattices with a scale of periodically dielectric constant comparable to a light wave length. They also have an ability to control both the speed of optical radiation in such structures [13] and the localization of electromagnetic waves [14]. Examples of these structures, such as iridoplasts in begonia [4], chloroplasts of *Selaginella erythropus* [5], bisonoplasts [15], giant chloroplasts [16], lamelloplasts [17], algae [18], epicuticular wax [6], and others [19], have been described in a number of studies.

The influence of a biophotonic crystal structure on the spectral characteristics and efficiency of photosynthesis are described in Reference [20]. Studies that have been carried out so far describe only strictures that have a biophotonic crystal structure on the surface layer [4]. In most natural objects, such a structure in chloroplasts is located under many layers, which have different size and parameters. This makes its direct measurements quite challenging. In these cases, valuable information can be provided by transmission electron microscopy (TEM).

One of the most sought-after plants in the world is wheat. It holds the second place in harvest volume and plays a huge role in the world economy. Moreover, it shows good growth index and stress resistance [21,22]. That is why we have chosen this plant as a sample of the structural arrangement of the photosystem. A huge number of publications are devoted to wheat, but studies of the morphology of grana stacks consisting of thylakoid membranes do not describe the angle of the electron beam aimed at the grana positioning. However, even several degrees deflection from a 90 angle can distort the grana dimension up to one hundred nanometers. In addition to that, thylakoid stacks can vary in average size depending on the type of a plant (wheat). Therefore, it was important for us to perform our own measurements so that we could be sure of the accuracy of the true grana size in a particular type of wheat used for our calculation model. When observing samples using TEM, it is always necessary to not only take a clear picture but also to determine whether a similar structure lies below it because they can overlap each other in the final photograph and distort the dimensions. As far as the study of stacks of thylakoids is concerned, their plane can be rotated relative to the incidence of the TEM beam; thus, the dimension of a granum may also be distorted. In this work, we determined the role of structural features of chloroplasts in the process of light collection. We constructed a model of a one-dimensional photonic crystal based on the sizes of thylakoid grana and the distances between them, while accounting for the absorption lines of chlorophylls. It was also important for us to determine the most accurate grana sizes when obtaining our own images of the wheat structure.

## 2. Materials and Methods

### 2.1. Electron Microscopy

At first, flag leaves of healthy average field wheat, Krasnoyarskaya 12, were collected at the ear stage. Then, they were fixed with 2.5% glutaraldehyde in phosphate buffer, followed by additional fixation with 1% $OsO_{4}$ (Sigma, St. Louis, MO, USA) in distilled water at room temperature. After that, the samples were dehydrated with ethanol and acetone and impregnated with a mixture of epoxy resins and araldite in a 4:1 ratio [23]. Impregnation and polymerization were performed as described in Reference [24]. A Leica EM UC7 ultramicrotome was used to obtain ultrathin sections, and a Hitachi HT 7700 transmission electron microscope was used to get digital images.

### 2.2. Optical Model of Chloroplast and Calculations

As refractive indices differ in each layer, the light passing through them is repeatedly re-reflected at boundaries. That is why waves in each layer move in opposite directions with amplitudes ($A_{i}$ and $B_{i}$), respectively.

The transfer-matrix method makes it possible to simplify the computer calculation of the amplitudes of stationary (settled in time) waves in each layer. Knowing $A_{i}$ and $B_{i}$, we are able to calculate $A_{i-1}$ and $B_{i-1}$. To do this, we need to know the refractive indices ($n_{i}$ and $n_{i-1}$), the layer thickness ($Z_{i}$), and the wave frequency.

So, we can write that ($A_{i-1}$, $B_{i-1}$) = $F_{n}$ ($A_{i}$, $B_{i}$, $Z_{i}$, $n_{i-1}$, $n_{i}$, ω).

The $F_{n}$ function is the same for every pair of layers. So, given the initial conditions ($A_{out}$ = 1, $B_{out}$ = 0), and using the $F_{n}$ function, it is possible in N + 1 cycles to find ($A_{0}$, $B_{0}$), i.e., the amplitudes of the incident and reflected waves.

Knowing that only an outgoing wave exists at the exit from the structure ($A_{out}$ = 1; $B_{out}$ = 0), and after performing numerical calculations, we can obtain an array of relative values of amplitudes in each of the PC layers. Thus, it is possible to find the distribution of the electromagnetic field in the layered structure and its transmission spectrum.

Transmission coefficient *T* [14] (prerequisite: the refractive indices of the media on both sides of the sample are the same):
(1)$$T=1-{\left|\frac{B_{0}}{A_{0}}\right|}^{2}.$$

To calculate the density of photon states, we used the formula obtained in Reference [25]:(2)$$\rho_{\omega }=\frac{\frac{1}{2L_{\Sigma }}\int_{0}^{L_{\Sigma }}\left[\epsilon_{\omega }\left(z\right){\left|E_{\omega }\right|}^{2}+\frac{c^{2}}{\omega^{2}}{\left|\frac{\partial E_{\omega }}{\partial z}\right|}^{2}\right]\partial z}{{c|E_{\omega }^{I}|}^{2}},$$ where $E_{\omega }$ is the amplitude of the electric component of the electromagnetic field, $E_{\omega }^{I}$ is the amplitude of the incident wave, $\epsilon_{\omega }\left(z\right)$ is the dielectric constant from the coordinate, and $L_{\Sigma }$ is the total thickness of the structure. The formula shows the ratio of energy in a layered medium to energy in vacuum at the same geometric length.

The density of photon states graph is a set of points. Each point shows the ratio from Formula (2) at different frequencies. In order to make the calculations while considering dispersion, we changed the real part of the refractive index in the area of the chlorophylls absorption lines a and b according to the method presented in Reference [26].

## 3. Results

Photographs of chloroplasts structure were obtained using transmission electron microscopy. Figure 1 presents sections of a wheat chloroplast. Periodic structures—grana that consist of densely packed thylakoids—usually have packages sized ∼300 nm and ∼100 nm (Figure 1a). The sizes have been estimated using a number of images. Thylakoids (including the lumen) are about 12 ± 2 nm wide (n = 15), and the stromal distance is 9 ± 1.5 nm. A series of images of this area made at different angles of inclination of the sample to the beam shows that these structures are located at different levels along the slice thickness. Changing the position of the sample according to the electron beam angle, we found the optimal position when the granum is located strictly perpendicular, so that its absolute dimensions can be observed without any distortion. The average distance between membranes is about 60 nm. Since the sample is about ∼150–200 nm thick, the thylakoids located obliquely to the beam have a smaller electron capture cross section than the layers perpendicular to the incident radiation. Therefore, they look blurry, and the periodic structure in them can hardly be noticed. Figure 1b shows a similar package of the same sample with a dimension of about 300 nm, but at a different angle. Grana stack with typical dimensions of about 140 nm can be seen in it.

Since thylakoids (and the distance in stacks between them) are extremely small individually, to make spectral activity noticeable in the visible range, it is better to consider a whole stack of thylakoids (granum) as a single layer with an average refractive index. Consequently, the stromal distance will be the second layer in a structure consisting of two periods, repeating one after another. Therefore, the period in such structures can be compared to the wavelength of visible light.

In Reference [27], a method for finding hidden ordering was proposed for quasi-ordered structures, which makes it possible to study them as photonic crystals. The authors consider order to be the foundation of “chaos”. They not only explained the silveriness of fish with the help of fractal geometry but also theoretically reproduced such reflectors. One of the oldest fractals, the Cantor set, was used to present a disordered structure.

Usually, the Cantor set is constructed step by step. A section defining the full optical path equal to the length of the observed periodic medium is divided into three equal parts. After that, the refractive index of its middle part is replaced with a different one, and the two parts on the edges go through the stage of division and replacement again. These steps can be repeated indefinitely.

This approach is convenient for approximating unknown specific values of periodic structures, but it can hardly be applied to biological ones. To adopt it to biological objects, generators of random deviations were added at each stage of crystal construction. The random deviation method was used and described in detail in Reference [27].

This method was used to select a periodic structure which the sizes of layers corresponding to the grana and stroma from the images we obtained before (Figure 2).

Taking into account the parameters specified in Figure 2 and using the transfer-matrix method, we calculated the transmission spectrum and the graph of the density of photon states (Figure 3a,b).

In this study, we considered normal incidence of light in a nonmagnetic medium (μ = 1), consisting of layers that have thickness $z_{N}$ and a refractive index $n_{N}$. A plane electromagnetic wave spreads along the $O_{z}$ axis. The amplitudes of waves (A and B) going in the right and left directions, respectively, in the previous layer depend on the same values in the current one [28].

The calculated transmission spectrum has a band gap in the visible range at a wavelength of 505–590 nm. Its graph of the density of photon states presents highest values at the edges of the band gap.

In the model that excludes dispersion, the selective reflection zone falls on the green color range.

Figure 3c,d present the calculation results including dispersion. The graphs show that the zone of selective reflection in the green range has been preserved. In addition to that, a new band gap has appeared in the red zone, the edges of which are located at the points of effective photosynthesis. We can observe some peaks of density of photon states at the edges of the band gaps. Thus, an increase in the probability of quantum processes occurs at these wavelengths. This spectral pattern is very stable if the period of the structure changes within up to 10%; with a further increase, the stop band in the green area disappears but remains in the red area. Even an increase in the density of photon states at peak values may be noticed.

Due to the movement of thylakoids, the size of a granum can vary by up to 30% depending on external conditions [29]. After the granum width was increased in our model by more than 10%, we obtained the case shown in Figure 4.

In the case presented in Figure 4c,d, for the frequency that corresponds to the highest density of photonic states, the graph of the module of amplitude of the electric component along the photonic crystal along the layers was calculated (Figure 5).

In Figure 5, the incident wave corresponds to 1. It can be observed that the amplitude is greater relative to the incident wave. So, a local amplification of the electromagnetic field exists at this frequency.

The first case (Figure 3) shows selective reflection in the green part of the spectrum, and, as a consequence, a decrease in the density of photon states in this wavelength range, with the exception of the stop band boundaries. In the second case (Figure 4), the density of photon states in the same parts of the spectrum changes slightly, but it is by a huge ratio higher than in the first case. This can happen more efficiently (up to an order of magnitude) in the presence of so-called “defect modes” arising when the dimensions of individual layers change.

## 4. Discussion

The data show that, for different lattice parameters without dispersion taken into account (Figure 3a,b and Figure 4a,b), the transmission spectrum and the density of photon states change slightly. However, when dispersion is considered (Figure 3c,d and Figure 4c,d), the density of photon states in the red area stays the same, while, in the green area, the whole band gap has almost disappeared. So, we can conclude that both amplification of the amplitude and its weakening at certain frequencies are possible in a layered medium, due to resonance phenomena. This explains why biological objects can show such great adaptation to external influence. The spectral bandwidth at which these effects are noticed can also vary depending on the geometric parameters. An interesting fact is that the band gap in the area of effective photosynthesis does not disappear in such structures, even with fifty percent disordering [28].

Schrödinger [30], in his studies, was fair to say that biological structures are “aperiodic crystals”. Thereby, he confirmed their high orderliness.

It can be seen from our results that the structure of a wheat chloroplast has an ordered structure, where the period is comparable to the wavelength of the visible range. A distinctive feature of wheat chloroplasts is the presence of ordered cylindrical disc-shaped grana. In such structures, the manifestation of strong interactions between exciton, Bragg, and lattice resonances is possible [8,20,26]. Figure 3 and Figure 4 show how the density of photon states increases in the area of effective photosynthesis. This is because the local amplitude is greater than the amplitude of the incident wave (local amplification).

According to Fermi’s golden rule, the rate of chemical reactions that occur under the influence of light is proportional to the density of photon states. In addition, we should keep in mind that the probability of a photochemical reaction depends on the local amplitude of the electric field. The relationship between photosynthesis and the electric field was proposed by Witt [31,32].

The efficiency of photosynthesis also depends on the mechanisms of intermolecular energy transfer. Kazanov, in his work [33], shows that there is a slowdown of light in photonic crystal structures, which promotes the energy capture by reaction center in any of the mechanisms of intermolecular energy transfer [3]. The strongest effect it may have on the mechanism of emission and absorption of quanta.

In addition to layered periodic structures, there are other photonic crystal formations in plants. For example, opal-like photonic crystal structures were found in the rainbow algae *Cystoseira tamariscifolia* [18]. Algae can form monodisperse spheres consisting of lipid microspheres and control the dimension of their pack, as well as change the pack structure from ordered to disordered, and vice versa. Lopez-Garcia also calculated the reflectance as a function of the opal lattice expansion, which showed that the expansion of the lattice leads to redshift of the center wavelength of the reflectance peak. The calculated absolute reflectivity does not decrease but, rather, increases due to the expansion of the lattice. As packs with opals inside are active in low light conditions, their role is to efficiently scatter light in low light conditions onto chloroplasts, which are otherwise in a photoprotective alignment. This adaptation may be a response to abrupt changes in lighting conditions in which this littoral alga lives. Provided that chlorophylls are also located at the boundaries of the internal spherical structures, a complete analogy with our study can be drawn.

In addition to layered periodic structures, there are other photonic crystal formations in plants. Helicoidal ordering is the most common form of chiral structures. The periodicity here occurs only along one coordinate, while the structure is homogeneous along the other two directions.

The effect that helicoidal structure has on the optical fields depends on the ratio of the incident light wavelength (λ) and the pitch of the spiral (P): λ<P-negative optical activity of the polarization rotation in the opposite direction of the helicoid twisting; λ>P-Positive optical activity, linearly polarized light tracks the spiraling of the helicoid; λ=P-equality of conservation in the frequency range $|\lambda -\sqrt{\varepsilon \mu }|\leq \delta$ for the circular polarization component of the incident wave. Its sign coincides with the sign of the twisting of the helicoid. In this case, selective dimensional reflection occurs. Circular Bragg diffraction appears.

We cannot help but admire how amazingly living objects can use the same method to carry out various processes. For example, iridoplasts were found in the epidermal cells of begonia leaves. Their optical properties were measured and modeled along with photonic crystals. The results obtained corresponded to the color of the plant, thereby describing not only the color of leaves but also the iridescence visible on them [4]. Masters, in his work [5], investigated bizonoplasts in S. Erythropus. These bizonoplasts are a unique form of chloroplasts and, at the same time, a one-dimensional photonic crystal. This structure results in enhanced reflection in the blue area of the visible spectrum. These bizonoplasts are very similar to iridoplasts in structure and have a selective reflection zone in the same area.

Some recent studies have shown that the epicuticular wax on the leaves of gray wheat and blue spruce is directly responsible for plant coloration due to its unique photonic crystal nanostructure [6]. The epicuticular wax of these plants consists of nanotubes and nanorods with an outer diameter of 140–160 nm. It is significant that plant cuticular waxes consist of a complex mixture of long-chain aliphatic compounds, which can be classified according to the type of functional groups, structure, and distribution of homologues [34]. In addition to that, various amounts of cyclic compounds are often observed. In Reference [34], it is reported that self-organized wax units can form anisotropic crystals. Due to its heterogeneous properties, epicuticular wax is able to not only reflect a certain part of visible light but also to absorb ultraviolet radiation and even convert it into visible light with the help of fluorescence. So, more and more examples of natural photonic crystals that play an important role in the life of plants have been discovered recently.

## 5. Conclusions

The study shows how effectively biophotonic crystals concentrate the radiation energy over distance and frequency. According to Fermi’s golden rule, this increases the probability of photochemical reactions.The mechanisms of changes in the density of photon states and the local electric field, that take place due to absorption lines, have been revealed.

## Figures and Tables

**Figure 1 plants-10-01967-f001:**
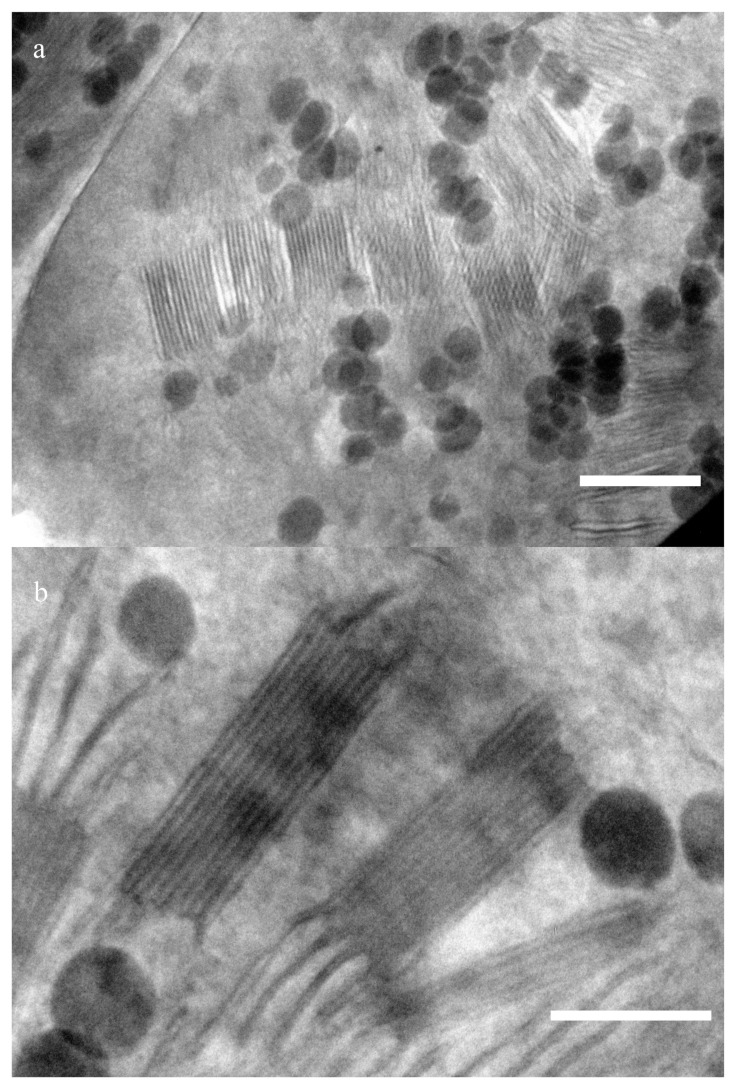
Transmission electron microscopy images of chloroplast in wheat samples: (**a**) 500 nm scale. (**b**) 200 nm scale.

**Figure 2 plants-10-01967-f002:**
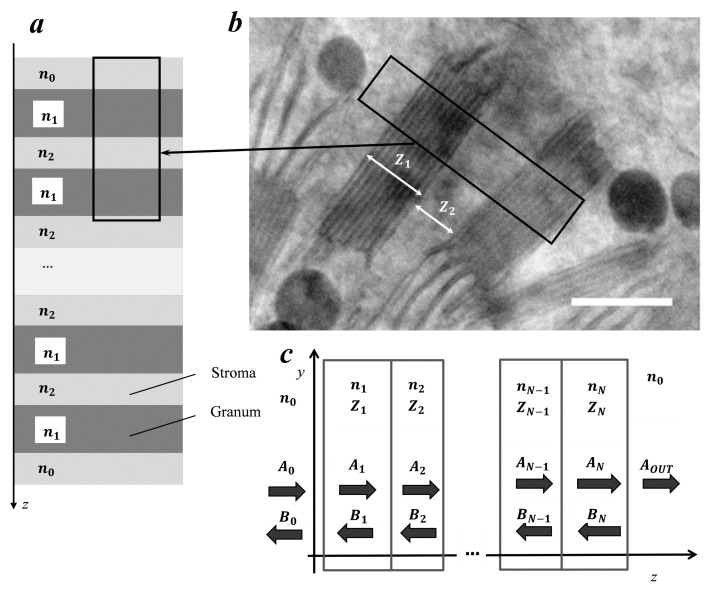
Simulation of a photonic-crystal layered structure with plane electromagnetic waves propagating along the *z*-axis. The black rectangle on the left corresponds to the right part of the image. Average granum thickness $Z_{1}$ = 120 nm. Average stromal distance between the grana $Z_{2}$ = 70 nm. Refractive indices: $n_{1}$ = 1.48 for granum; $n_{0}$ = $n_{2}$ = 1.33 for stroma. (**a**) Visual model representing the photonic crystal used in our calculations. (**b**) Part of the image taken from Figure 1b. (**c**) Distribution of electromagnetic wave amplitudes along a photonic crystal.

**Figure 3 plants-10-01967-f003:**
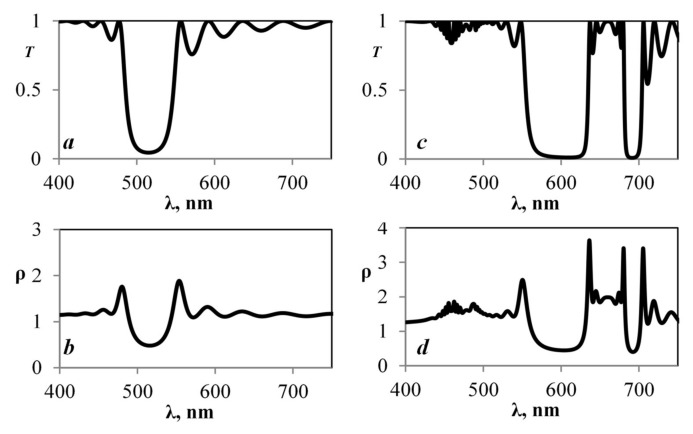
Design graphs for the periodic structure obtained with the help of the Cantor method. (**a**) Transmission spectrum excluding dispersion. (**b**) Graph of the density of photon states excluding dispersion. (**c**) Transmission spectrum including dispersion. (**d**) Graph of the density of photon states including dispersion.

**Figure 4 plants-10-01967-f004:**
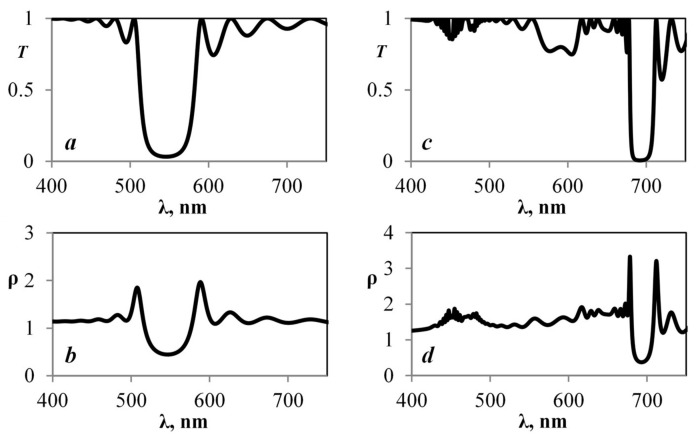
Graphs for the periodic structure obtained using the Cantor method. Period increased by 10%. (**a**) Transmission spectrum excluding dispersion. (**b**) Graph of the density of photon states excluding dispersion. (**c**) Transmission spectrum including dispersion. (**d**) Graph of the density of photon states including dispersion.

**Figure 5 plants-10-01967-f005:**
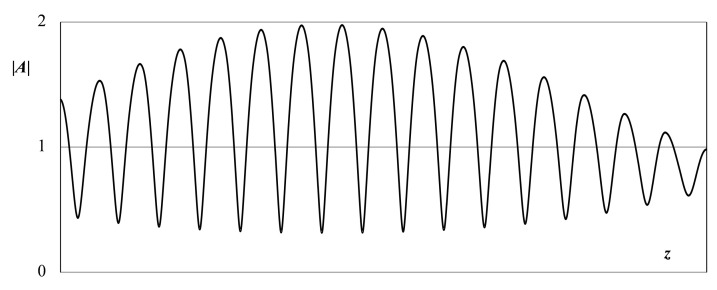
Amplitude of the electric component of electromagnetic field in a periodic structure obtained using the Cantor method and a period increased by 10% at λ = 680 nm.

## Data Availability

Not applicable.

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
