# Peer review of "The Role of Periodic Structures in Light Harvesting"

_plants, 2021, doi:10.3390/plants10091967_

Round 1

Reviewer 1 Report

The manuscript of Bukhanov et al “The role of periodic structures in light harvesting” utilizes a method for determining the hidden ordered structure in thylakoid membranes by using the formalism of transfer matrix method on recorded transmission spectra and  transmission electron microscopic images. The topic is relevant to plant physiology research since there are a number of evidences that multilevel ordering in photosynthetic membranes plays crucial part in the optimization of the photosynthetic process. However, a major drawback of the manuscript is the unclear relevance of the obtained results for the regulation of the photosynthetic process.

From the Abstract the reader expects to learn about the hidden ordered structure of LHCII but from the Results and Discussion sections this is information is not obtained. The presented results as far as I understand concern the ordering of grana membranes and not LHCII in particular.

In the Introduction the authors explain the different structural elements (including multilayered structures) that might affect the color of biological objects and clarify the difficulty of studied performing direct measurements on LHCII due to its localization deep in the plant cell and chloroplasts. The authors claim that the manuscript defines “the role of structural features of chloroplasts in the process of light collection”, but in the Results and Discussion sections it does not become clear what exactly this role is and what is the novel information obtained. For a plant biologist with no experience with the Transfer-matrix method, it is not clear what additional information it provides, taking into account that there are high resolution microscopic and spectroscopic techniques that provide detailed information on the ordering of the photosynthetic complexes within the thylakoid membrane. Furthermore, time-resolved fluorescence and absorption techniques provide picosecond (and even faster) resolution of the processes associated with light-harvesting in photosynthetic organisms and structural methods provide atomistic details on the pigments and proteins involved in this process. I believe its crucial for the authors to clarify this point if they aim at publishing in Plants as well as to invest more effort to make the paper understandable for plant biologists.

Materials and Methods does not contain any information on the method of modelling of the imaged structures and hence how the transmission spectra and the graph of the density of photon states are obtained. Also a clear explanation of what is the biological/physiological/structural meaning of those parameters would be very helpful for the readers with biological background.

Caption Figure 1 – I guess authors mean chloroplasts instead of “chloroplast cells”.

How are the refractive indices for grana and inter-grana distance used in the computational model obtained?

Figure 2 – What is the “stable layered structure” characterized in the figure? Is it the grana structure?

For what kind of light quality the calculations are performed and how do they depend on the incident wavelength and light intensity?

Row 127-128 “This spectral pattern is very stable when the parameters of the physical structure change,” – please clarify which physical parameters have you tested and show relevant data

rows 134-140 : “The first case (Fig. 2) shows selective reflection in the green part of the spectrum,

135 and, as a consequence, a decrease in the density of photon states in this wavelength

136 range, with the exception of the stop band boundaries. In the second case (Fig. 3), the

137 density of photon states in the same parts of the spectrum changes slightly, but it is by a

138 huge ratio higher than in the first case. This can happen more efficiently (up to an order

139 of magnitude) in the presence of so-called “defect modes” arising when the dimensions

140 of individual layers change.” In Figures 2 and 3 the authors compare the density of photon states in granas with different size but it is not clear if the difference is in lateral or vertical direction, or in the volume of the granas. Again, what is the physiological consequence of the findings presented in Figs. 2 and 3 and what is the benefit of applying this theoretical method in comparison with many other experimental techniques that provide information on the chloroplast photochemical reactions and structural organization?

Discussion contains too much information that is not relevant to the photosynthetic organisms, as well as a lot of information that concerns photosynthetic organisms but it is not clear how it is related to the research question.

The effect that helicoidal structure on the optical fields is detailed but the authors do not specify why this is important for the current research – my guess is because of the helical model of structural organization of grana membranes, as shown by some authors but this is by far not clear for the reader. Relevant references for the detailed grana structure might help to make this more clear.

Conclusions

258“1. The study shows how effectively biophotonic crystals concentrate the radiation

259 energy over distance and frequency. According to Fermi’s golden rule, this increases the

260 probability of photochemical reactions.” – What is the new information obtained in this work? It is well known that the light-harvesting process is related to concentration and direction of light energy from the antennae complexes towards the reaction centers and many structural and kinetic details are well determined.

261 2. The mechanisms of changes in the density of photon states and the local electric field,

262 that take place due to absorption lines, have been revealed.- Clarify what are the exact mechanisms that you have in mind?

264 3. Describing resonance phenomena in a reconfigurable layered medium makes it possi

264ble to explain how biological objects can adapt so greatly to the conditions of external

265 radiation. Both strengthening of the local amplitude of electromagnetic radiation (for

266 low illumination), and its weakening (to prevent the destruction of organic components

267 from excessive light) are possible. The spectral bandwidth at which these effects will be

268 observed can also vary depending on geometric parameters of the resonator systems. – This text is more appropriate for Discussion and not for Conclusion section since in the manuscript no real experiments with low/high light are performed. The only thing that is modelled is the increase in grana and it is also not confirmed with experiment in which grana with different sizes are observed.

In summary, my recommendation is “major revision” and strong advice to the authors to put the manuscript in a plant physiology context so that the readers of Plants could really understand and benefit from the applied theoretical modelling.

Author Response

Thank you for the objective review and opportunity to revise our paper. All edits are highlighted in red in file in attachment.

Point 1: The manuscript of Bukhanov et al “The role of periodic structures in light harvesting” utilizes a method for determining the hidden ordered structure in thylakoid membranes by using the formalism of transfer matrix method on recorded transmission spectra and  transmission electron microscopic images. The topic is relevant to plant physiology research since there are a number of evidences that multilevel ordering in photosynthetic membranes plays crucial part in the optimization of the photosynthetic process. However, a major drawback of the manuscript is the unclear relevance of the obtained results for the regulation of the photosynthetic process.   Response 1: Figures 2 and 3 show an increase in the density of photon states in the area of ​​effective photosynthesis. In accordance with the Fermi golden rule, this indicates an increase in the efficiency of the photosynthesis process. We know that light is slowed down in photonic crystal structures. This helps to transfer energy to the reaction center.   Point 2: From the Abstract the reader expects to learn about the hidden ordered structure of LHCII but from the Results and Discussion sections this is information is not obtained. The presented results as far as I understand concern the ordering of grana membranes and not LHCII in particular.
  Response 2: We agree with the comment on the structure of the chloroplast that we studied. We removed the parts of the text describing LHCII.  

Point 3: In the Introduction the authors explain the different structural elements (including multilayered structures) that might affect the color of biological objects and clarify the difficulty of studied performing direct measurements on LHCII due to its localization deep in the plant cell and chloroplasts. The authors claim that the manuscript defines “the role of structural features of chloroplasts in the process of light collection”, but in the Results and Discussion sections it does not become clear what exactly this role is and what is the novel information obtained. For a plant biologist with no experience with the Transfer-matrix method, it is not clear what additional information it provides, taking into account that there are high resolution microscopic and spectroscopic techniques that provide detailed information on the ordering of the photosynthetic complexes within the thylakoid membrane. Furthermore, time-resolved fluorescence and absorption techniques provide picosecond (and even faster) resolution of the processes associated with light-harvesting in photosynthetic organisms and structural methods provide atomistic details on the pigments and proteins involved in this process. I believe its crucial for the authors to clarify this point if they aim at publishing in Plants as well as to invest more effort to make the paper understandable for plant biologists.

Response 3: The research shows that an ordered structure with a period comparable to the wavelength of light in the visible range exists in chloroplasts. In such structures, exciton and Bragg resonances may show strong interaction. Due to this, local frequency and spatial amplifications of the electromagnetic field amplitude appear, and the PC densities change.

Point 4: Materials and Methods does not contain any information on the method of modelling of the imaged structures and hence how the transmission spectra and the graph of the density of photon states are obtained. Also a clear explanation of what is the biological/physiological/structural meaning of those parameters would be very helpful for the readers with biological background.

Response 4: We agree with this point. We have expanded this part in methods and results.

Point 5: Caption Figure 1 – I guess authors mean chloroplasts instead of “chloroplast cells”.

Response 5: Corrected, thank you.

Point 6: How are the refractive indices for grana and inter-grana distance used in the computational model obtained?

Response 6: We agree with the comment. Formulas are given in methods and shown in new Figure 2.

Point 7: Figure 2 – What is the “stable layered structure” characterized in the figure? Is it the grana structure?

Response 7: Corrected and explained in more detail. "A stable layered structure" is a photonic crystal that does not change its optical characteristics with a slight distortion of its physical parameters. Granum is one layer of this crystal.

Point 8: For what kind of light quality the calculations are performed and how do they depend on the incident wavelength and light intensity?

Response 8: This is a one-dimensional case. The wave falls at a 90 degree angle. We did not consider the dependence on the incident wave intensity. Calculations are made for the entire visible spectrum.

Point 9: Row 127-128 “This spectral pattern is very stable when the parameters of the physical structure change,” – please clarify which physical parameters have you tested and show relevant data

Response 9: Layer thicknesses and refractive indices were tested. The data are shown in the graph.

Point 10: rows 134-140 : “The first case (Fig. 2) shows selective reflection in the green part of the spectrum,

135 and, as a consequence, a decrease in the density of photon states in this wavelength

136 range, with the exception of the stop band boundaries. In the second case (Fig. 3), the

137 density of photon states in the same parts of the spectrum changes slightly, but it is by a

138 huge ratio higher than in the first case. This can happen more efficiently (up to an order

139 of magnitude) in the presence of so-called “defect modes” arising when the dimensions

140 of individual layers change.” In Figures 2 and 3 the authors compare the density of photon states in granas with different size but it is not clear if the difference is in lateral or vertical direction, or in the volume of the granas. Again, what is the physiological consequence of the findings presented in Figs. 2 and 3 and what is the benefit of applying this theoretical method in comparison with many other experimental techniques that provide information on the chloroplast photochemical reactions and structural organization?

Response 10: We agree with the comment and attempted to give a more detailed explanation. In this work, for the first time, we studied how grana ordering in chloroplasts affect the features of light distribution in a plant leaf.

Point 11: Discussion contains too much information that is not relevant to the photosynthetic organisms, as well as a lot of information that concerns photosynthetic organisms but it is not clear how it is related to the research question.

Response 11: We agree with the comment and applied all the necessary corrections.

Point 12: The effect that helicoidal structure on the optical fields is detailed but the authors do not specify why this is important for the current research – my guess is because of the helical model of structural organization of grana membranes, as shown by some authors but this is by far not clear for the reader. Relevant references for the detailed grana structure might help to make this more clear.

Response 12: We stated that not only layered structures can have properties of photonic crystals. Helicoidal structures which also exist in living nature can possess these properties too.

Conclusions

Point 13: 258“1. The study shows how effectively biophotonic crystals concentrate the radiation

259 energy over distance and frequency. According to Fermi’s golden rule, this increases the

260 probability of photochemical reactions.” – What is the new information obtained in this work? It is well known that the light-harvesting process is related to concentration and direction of light energy from the antennae complexes towards the reaction centers and many structural and kinetic details are well determined.

Response 13: In our work we pay attention to the large role played by resonant interactions. They lead to new effects of amplification of local frequency and spatial amplifications of the electromagnetic field amplitude. This, in turn, influences the efficiency of photosynthesis.

Point 14: 261 2. The mechanisms of changes in the density of photon states and the local electric field,

262 that take place due to absorption lines, have been revealed.- Clarify what are the exact mechanisms that you have in mind?

Response 14: Resonant interaction and dispersion.

Point 15: 264 3. Describing resonance phenomena in a reconfigurable layered medium makes it possi

264ble to explain how biological objects can adapt so greatly to the conditions of external

265 radiation. Both strengthening of the local amplitude of electromagnetic radiation (for

266 low illumination), and its weakening (to prevent the destruction of organic components

267 from excessive light) are possible. The spectral bandwidth at which these effects will be

268 observed can also vary depending on geometric parameters of the resonator systems. – This text is more appropriate for Discussion and not for Conclusion section since in the manuscript no real experiments with low/high light are performed. The only thing that is modelled is the increase in grana and it is also not confirmed with experiment in which grana with different sizes are observed.

Response 15: We agree with the comment. Alterations made.

Reviewer 2 Report

The authors investigated the optical properties of thylakoids in wheat. Very interesting approach. Several findigns are interesting, however the discussio, especially regarding helical strcutres and algae/cyanobacteria are missleading and with some error. THis needs to be corrected.

Some minor things below.

Figure description unclear, please describe in detail the differences between fig 2 and 3

Please add some details regarding the calculations in the material and method part.

In the discussion, the helical structure part is not essential and can be shorter.

Please put all species names in italics;  also vice versa etc.;

60 nm or 60nm please correct

Line 72: sigma...

Line 168: Reference?

Line 185: Papenfuss?

Line 817: Reference unclear

Line 203: Cyanobacteria and red algae contain chlorophyll a and phycobilins. Please rewrite paragraph

Author Response

We are thankful for your review and agree with every comment. The text was corrected where needed. All edits are highlighted in red in the file in the attachment.

Round 2

Reviewer 1 Report

The authors have significantly improved the manuscript.